# Postpartum long-acting reversible contraceptives use in sub-Saharan Africa. Evidence from recent demographic and health surveys data

**Desale Bihonegn Asmamaw**[1] *, **Tadele Biresaw Belachew**[2], **Samrawit Mihret Fetene**[2], **Banchlay Addis**[2], **Tsegaw Amare**[2], **Atitegeb Abera Kidie**[3], **Abel Endawkie**[4], **Alebachew Ferede Zegeye**[5], **Tadesse Tarik Tamir**[6], **Sisay Maru Wubante**[7], **Elsa Awoke Fentie**[1], **Wubshet Debebe Negash**[2]

1 Department of Reproductive Health, Institute of Public Health, College of Medicine and Health Sciences, University of Gondar, Gondar, Ethiopia, 2 Department of Health Systems and Policy, Institute of Public Health, College of Medicine and Health Sciences, University of Gondar, Gondar, Ethiopia, 3 School of Public Health, College of Health Science, Woldia University, Woldia, Ethiopia, 4 Department of Epidemiology and Biostatistics, School of Public Health, College of Medicine and Health Science, Wollo University, Dese, Ethiopia, 5 Department of Medical Nursing, School of Nursing, College of Medicine and Health Sciences, University of Gondar, Gondar, Ethiopia, 6 Department of pediatric and child health nursing, school of nursing, College of medicine and health sciences, University of Gondar, Gondar, Ethiopia, 7 Department of Health Informatics, Institute of Public Health, College of Medicine and Health Sciences, University of Gondar, Gondar, Ethiopia

* desalebihonegn1988@gmail.com

## Abstract

### Background

In developing countries, most women want to avoid pregnancy for two years after giving birth. However, 70% do not use contraceptives during this time. Unintended pregnancies may occur for couples who delay contraceptive use during the postpartum period. The most effective form of contraceptive methods for postpartum women is long-acting reversible contraceptive (LARC). Therefore, this study aimed to assess long-acting reversible contraceptive use and associated factors among postpartum women in Sub-Saharan Africa.

### Methods

Secondary data analysis was performed using the recent Demographic and Health Surveys (DHS). Stata version 14 was used to analyze the data. A multilevel mixed-effect logistic regression model was used to identify factors associated with long-acting reversible contraceptive use. Variables with a p-value < 0.05 in the multilevel mixed-effect logistic regression model were declared significant factors associated with long-acting reversible contraceptives.

### Results

The magnitude of long-acting reversible contraceptive use among postpartum women was 12.6% (95% CI: 12.3, 12.8). Women primary (aOR = 1.51; 95% CI: 1.41, 1.63) and

**Data Availability Statement:** All relevant data are within the paper and its Supporting Information files.

**Funding:** The authors received no specific funding for this work.

**Competing interests:** The authors have declared that no competing interests exist.

secondary education (aOR = 1.62; 95% CI: 1.32, 1.71), media exposure (aOR = 1.73; 95% CI: 1.51, 1.85), place of delivery (aOR = 1.54; 95% CI: 1.43, 1.67), number of ANC visit; 1–3 (aOR = 2.62; 95% CI: 2.31, 2.83) and ≥4 (aOR = 3.22; 95% CI: 2.93, 3.57), received PNC (aOR = 1.34; 95%CI: 1.13, 1.58), and income level; low middle income (aOR = 2.41; 95% CI: 2.11, 2.88) and upper middle income (aOR = 1.83; 95% CI: 1.56, 1.24) were significantly associated with long-acting reversible contractive use.

## Conclusion

Nearly one in 10 postpartum women used long-acting reversible contraceptives. Hence, we suggest that the concerned bodies should promote family planning messages in mass media and give the well-documented benefits of postpartum long-acting contraceptive use. Promote the integration of postpartum LARC methods into maternal health care services and give better attention to postpartum women living in low-income countries and uneducated women.

## Introduction

The World Health Organization (WHO) recognizes family planning (FP) as an essential reproductive health intervention that is provided during the antenatal period, immediately after delivery, and during the first year of the postpartum period [1, 2]. Postpartum family planning (PPFP) is to prevent closely spaced and unintended pregnancies during the first year after childbirth [2]. Globally, the contribution of unintended pregnancy to maternal and child morbidity and mortality is significant [3]. Postpartum women are at risk of unintended pregnancy [4, 5]. More than 50% of postpartum women reported having had unprotected sexual intercourse before the routine 6-week postpartum visit [3, 6].

Currently, WHO and the United States Agency for International Development (USAID) recommend a minimal live birth interval of 2 years to reduce the risks of abortion, miscarriage, and stillbirth. PPFP is a primary strategy for reducing unintended pregnancy, optimizing birth spacing, and reducing maternal mortality and morbidity [7]. The most effective form of FP is long-acting reversible contraception (LARC) [6]. LARC is user-independent, and once the device is inserted, the woman does not need to take any action to support the ongoing effective utilization of contraception [8]. It is more effective in preventing unintended pregnancy, has higher continuation rates than shorter-acting methods, and the return of fertility is rapid when it is removed [8].

By 2030, the Sustainable Development Goals (SDGs) aim to reduce the global maternal mortality ratio (MMR) to less than 70 per 100,000 live births, with no country exceeding twice the global MMR (140 per 100,000 live births) [9]. Each year, 2.9 million newborns and 265,000 mothers die from complications that occur during pregnancy, childbirth, and the postpartum period. Of these, more than half occur in Sub-Saharan Africa (SSA) [10]. In order to reduce maternal and child deaths, many strategies have been implemented [11]. Family planning is one of the four pillars of safe motherhood initiatives and aims at preventing and reducing maternal mortality rates in developing countries, including SSA, through the control of the number of births a woman may have [12]. Any maternal and neonatal deaths are largely preventable in developing countries by expanding the use of postpartum LARC methods [5, 13, 14].

Even though the postpartum period is the best opportunity for LARC insertion, limited studies have documented LARC method use and associated factors [5, 6]. Furthermore, few studies have been conducted on LARC methods [4, 15, 16], however, majority of these focused on adolescent and reproductive-age women. As per our search of the literatures, no study has been conducted to investigate postpartum LARC use and related factors based on the pooled Demographic and Health Surveys (DHS) data. Investigating the magnitude of LARC use and its associated factors among postpartum women in SSA is crucial to assessing cross-national disparities. Besides, the study was able to detect the true effects of variables because it was based on pooled DHS data from the SSA. In order to improve maternal and child health, program planners will use the studies' results to allocate resources. Therefore, the aim of this study is to assess long-acting reversible contraceptive use and associated factors among postpartum women in SSA.

## Methods

### Study area, data source, and study design

In this study, we used secondary data from the SSA DHS from 2014 to 2019/2020. These datasets were appended together to investigate LARC use and associated factors among postpartum women in SSA.

We obtained the data from the official database of the DHS program, which can be accessed at http://www.dhsprogram.com. DHSs are nationally representative surveys that provide data that is comparable across countries for monitoring and impact evaluation indicators in the areas of population, health, and nutrition. In the DHS, households are sampled in two stages: enumeration areas (EAs) are selected (first stage), and samples are then drawn from every EA selected (second stage) [17]. We used DHS surveys done in 23 SSA countries and a total weighted sample of 78591 postpartum women who gave birth in the last one year before the surveys was included in the current study (Table 1).

### Variables of the study

**Dependent variable.** The outcome variable for this study was postpartum LARC use. LARC include intrauterine device (IUCD) and Implants. It was categorized as 1 (yes) if postpartum women used one of the above methods, and other wise 0 (no).

**Independent variables.** According to literature reviews, several independent variables were incorporated at the level of individuals and communities. Age of the women (15–24, 25–34, and 35–49), sex of child the last child (male, female), women's education (no formal education, primary education, and secondary education and above ), wanted last child (wanted then, wanted later, and wanted no more), number of ANC (0, 1–3, and ≥4), received PNC (yes, no), (place of delivery (home, health institutions) were considered as individual-level variables [6, 18–22]. Consumer goods like televisions, bicycles, and cars were considered in calculating the household wealth index. The household wealth index was calculated using roof, floor, and toilet materials. Through PCA, the wealth index is categorized into five wealth quartiles (poorest, poor, medium, richer, and richest) based on household asset data [19, 23]. Media exposure was coded as yes if the women read the newspaper, listen to the radio, or watch television at least once a week, and no if not [24].

Of the community level factors, residence (rural, urban) were directly accessed from the Ethiopian Demographic and Health Survey (EDHS) data set. Community-level independent variable was constructed based on individual-level characteristics aggregated at the community (cluster) level, such as poverty, media exposure, and education. A histogram was used to check the distribution of proportion values computed for each community and categorize them as high or low.

**Table 1. Sample size for postpartum long-acting reversible contraceptive use (sub-Saharan Africa, 2023).**

| Countries | Year of survey | Weighted sample (n) | Weighted sample (%) |
|---|---|---|---|
| Burundi | 2016/17 | 3443 | 4.4 |
| Ethiopia | 2016 | 3555 | 4.5 |
| Kenya | 2014 | 8735 | 11.1 |
| Malawi | 2015/16 | 8204 | 10.4 |
| Rwanda | 2019/20 | 3732 | 4.8 |
| Tanzania | 2015/16 | 3275 | 4.2 |
| Uganda | 2016 | 4181 | 5.3 |
| Zambia | 2018 | 3878 | 4.9 |
| Zimbabwe | 2015 | 2881 | 3.7 |
| Angola | 2015/16 | 3235 | 4.1 |
| Cameroon | 2018 | 2218 | 2.8 |
| Chad | 2014/15 | 2600 | 3.3 |
| Benin | 2017/18 | 3095 | 3.9 |
| Gambia | 2019/20 | 1955 | 2.5 |
| Ghana | 2014 | 2066 | 2.6 |
| Guinea | 2018 | 1706 | 2.2 |
| Liberia | 2019/20 | 2338 | 3.0 |
| Mali | 2018 | 1814 | 2.3 |
| Nigeria | 2018 | 6050 | 7.7 |
| Senegal | 2019 | 1701 | 2.2 |
| Sierra Leone | 2019 | 3832 | 4.9 |
| Lesotho | 2014 | 1863 | 2.4 |
| South Africa | 2016 | 2234 | 2.8 |
| Total sample size | | 78591 | 100 |

For categorization, the median value was used because the aggregate variable was not normally distributed [19, 25]. Based on the World Bank List of Economies classification since 2019, the countries' income levels were classified as low-income, lower middle income, and upper-middle-income. Based on Gross National Income per capita (GNI), countries income were calculated for countries with low income of $1025 or lower, lower middle income of $1026–3995, upper middle income of $3996–12,375, and high income of $12,375 or more [26].

## Data analysis

Stata version 14 statistical software was used to extract, code, and analyze data. To obtain reliable estimates and standard errors, all frequency distributions were weighted using the weight command in Stata (v005/1000000) before data analysis. The first approach involved the use of percentages to describe the LARC among postpartum women in SSA. This was followed by the distribution of LARC across the individual and community level factors. Pearson's chi-square test of independence ($X^2$) was used to assess the significance of the association between each independent variable and the LARC at a p-value of $< 0.05$. Finally, multilevel binary logistic regression analysis was done to assess the association between LARC and the individual and community-level factors. As a result of the EDHS data being hierarchical, a traditional logistic regression model cannot be used. As a result, postpartum women were nested within households, and households were nested within clusters. They may share similar characteristics within the cluster. As a result, multilevel binary logistic regression must take into account cluster variability.

Intra-class correlation coefficient (ICC) and proportional change in variance (PCV) were computed to measure the variation between clusters, and model fitness was done using the deviance (-2LLR). Moreover, multicollinearity was tested using the variance inflation factor (VIF), and we got a VIF of less than five for each independent variable with a mean VIF of 1.43, consequently, there was no significant multicollinearity among the independent variables. The null model (without independent variables), mode I (containing only individual-level factors), mode II (containing only community-level factors), and model III (containing both individual and community-level factors) were fitted after selecting variables for multivariable multilevel analysis. In the multivariable model, variables with an adjusted odds ratio (aOR) with a 95% confidence interval (CI) and a p-value of < 0.05 were considered significantly associated factors of LARC.

## Ethical approval

This study is a secondary data analysis of the DHS. In this study, the Macro International Institutional Review Board in Calverton, United States of America, as well as the National Ethical Review Committees in Burundi, Ethiopia, Kenya, Malawi, Rwanda, Tanzania, Uganda, Zambia, Zimbabwe, Angola, Cameroon, Chad, Benin, Gambia, Ghana, Guinea, Liberia, Mali, Nigeria, Senegal, Sierra Leone, Lesotho, and South Africa, approved the surveys. Informed consent was obtained from all participants before participation and all data was collected in a confidential manner. To use the survey data, MEASURE DHS provided us with raw data and written consent. There was no sharing or passing of the dataset to other bodies and it was kept confidential. The study is not an experimental study. All the methods were conducted according to the Helsinki declarations. More details regarding DHS data and ethical standards are available online at (http://www.dhsprogram.com).

## Results

In this study, the overall LARC use among postpartum women in SSA was 12.6% (95% CI: 12.3, 12.8), from all countries, Rwanda and Angola accounted for the highest (38.2%) and the lowest (1.5%) LARC use, respectively (Fig 1).

### Distribution of LARC across the individual and community level factors

Table 1 shows the results on the distribution of LARC use across the individual and community-level factors in SSA. The results indicate that LARC use was high among postpartum women aged 25–34 (15.2%), those with secondary education and above (14.7%), those who had media exposure (13.6%), and postpartum women in the rich wealth quintile (13.6%). Postpartum women also used LARC if they had institutional delivery (14%), those who had ≥ 4 ANC visits (13.5%), those who had received PNC (13.2%), those who had received family planning counseling within the last 12 months (14.4%), those who had 1–3 living children (15%), and those who had no wanted the last child (15%). There were also variations in the proportion of LARC use across the various community factors: lived in low-income countries (13.7%), lived in communities with high literacy (13.2%), lived in communities with high media exposure (13.0%), lived in communities with low poverty (12.9%), and lived in upper-middle-income countries (13.7%) (Table 2).

### Random effects and model fitness

LARC use varied significantly across clusters. In the baseline model without an independent variable, 41.6% of the variance in the LARC use was explained by the variation in

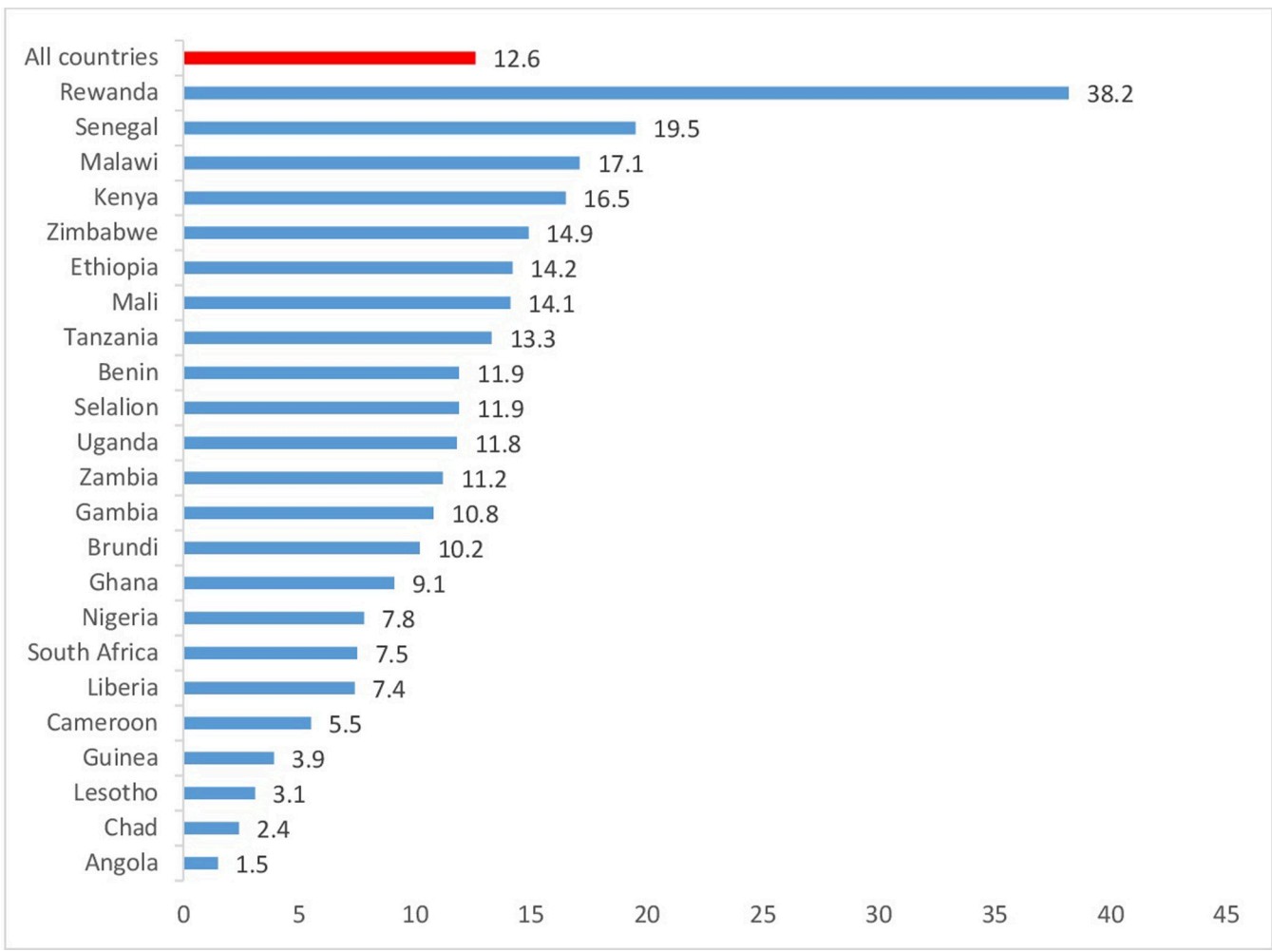

**Fig 1. Postpartum long-acting reversible contraceptive use (sub-Saharan Africa, 2023).**

characteristics between clusters (ICC = 0.416). The within-cluster variation was reduced to 34.5% in model 3, which included both individual and community factors. The variance in LARC use could therefore be explained by cluster differences. Additionally, the model with the lowest deviance value (model III) was found to be the best-fitting model (Table 3).

## Factors of LARC use in SSA

In the final model (model III) after adjusting for individual and community level factors, education, media exposure, place of delivery, number of ANC visit, PNC visit, and income level were significantly associated with LARC use among postpartum women.

Accordingly, the odds of LARC use among postpartum women who had primary and secondary education and above were 1.51 (aOR = 1.51; 95% CI: 1.41, 1.63) and 1.62 (aOR = 1.62; 95% CI: 1.32, 1.71) times higher than those who had no formal education, respectively. Postpartum women who had media exposure had 1.73 (aOR = 1.73; 95% CI: 1.51, 1.85) times more odds of using LARC than those who had no media exposure. The odds of LARC use among postpartum women who gave birth at health institutions (aOR = 54, 95% CI: 1.43, 1.67) were higher than those of postpartum women who gave birth at home. The odds of LARC use were

**Table 2. Distribution of long-acting reversible contraceptive use across individual and community level factors among postpartum women (sub-Saharan Africa, 2023).**

| Variables | Weighted N | Weighted % | LARC | $X^2$ (p-value) |
|---|---|---|---|---|
| Age | | | | 328.7 ($< 0.001$) |
| 15–24 | 24777 | 31.5 | 10.3 | |
| 25–34 | 31096 | 39.6 | 15.2 | |
| 35–49 | 22718 | 28.9 | 11.5 | |
| Sex of the last child | | | | 0.17 ($>0.05$) |
| Male | 40017 | 50.9 | 12.5 | |
| Female | 38574 | 49.1 | 12.6 | |
| Educational status | | | | 432.1 ($< 0.001$) |
| No formal education | 20138 | 25.6 | 9.1 | |
| Primary education | 29132 | 37.1 | 12.9 | |
| Secondary and above | 29321 | 37.3 | 14.7 | |
| Household wealth | | | | 142.9 ($< 0.001$) |
| Poor | 28760 | 36.6 | 11.0 | |
| Middle | 15396 | 19.6 | 13.2 | |
| Rich | 34435 | 43.8 | 13.6 | |
| Media exposure | | | | 386.5 ($< 0.001$) |
| No | 23678 | 30.1 | 10.1 | |
| Yes | 54913 | 69.9 | 13.6 | |
| Place of delivery | | | | 531.3 ($< 0.001$) |
| Home | 19255 | 24.5 | 8.4 | |
| Health institution | 59336 | 75.5 | 14.0 | |
| Number of ANC visits | | | | 361.5 ($< 0.05$) |
| No | 7244 | 9.2 | 6.4 | |
| 1–3 | 23171 | 29.5 | 13.1 | |
| $\geq 4$ | 48176 | 61.3 | 13.5 | |
| Received PNC | | | | 14.8 ($<0.001$) |
| No | 48510 | 61.7 | 12.2 | |
| Yes | 30080 | 38.3 | 13.2 | |
| Discuss with health professionals with the last 12 months | | | | 160.9 ($<0.001$) |
| No | 53770 | 68.4 | 11.8 | |
| Yes | 24821 | 31.6 | 14.4 | |
| Number of living children | | | | 351.7 ($<0.001$) |
| 0–1 | 27598 | 35.1 | 10.1 | |
| 1–4 | 33261 | 42.3 | 15.0 | |
| $\geq 5$ | 17731 | 22.6 | 12.0 | |
| Wanted last child | | | | 94.2 ($<0.001$) |
| Wanted then | 49504 | 63.0 | 11.6 | |
| Wanted later | 17177 | 21.9 | 13.6 | |
| Wanted no more | 11910 | 15.2 | 15.0 | |
| Resident | | | | 0.73 ($>0.05$) |
| Rural | 48592 | 61.8 | 12.5 | |
| Urban | 29998 | 38.2 | 12.6 | |
| Community level poverty | | | | 5.5 ($< 0.05$) |
| Low | 40842 | 52.0 | 12.9 | |
| High | 37749 | 48.0 | 12.2 | |
| Community-level media exposure | | | | 5.1 ($< 0.05$) |

*(Continued)*

**Table 2.** (Continued)

| Variables | Weighted N | Weighted % | LARC | $X^2$ (p-value) |
|---|---|---|---|---|
| Low | 36418 | 46.3 | 12.1 | |
| High | 42173 | 53.7 | 13.0 | |
| Community level education | | | | 13.7 (<0.001) |
| Low | 36802 | 46.8 | 12.1 | |
| High | 41789 | 53.2 | 13.2 | |
| Income level | | | | 244.6 (<0.001) |
| Low income | 43457 | 55.3 | 7.5 | |
| Lower middle income | 32900 | 41.9 | 11.5 | |
| Upper middle income | 2234 | 2.8 | 13.7 | |

2.41 (aOR = 2.41; 95% CI: 2.11, 2.88) and 1.83 (aOR = 1.83; 95% CI: 1.56, 1.24) times higher among postpartum women who lived in lower-middle-income and upper-middle-income households than their counterparts, respectively (Table 3).

## Discussion

The aim of this study was to determine LARC use and identify associated factors among postpartum women in SSA. This study found that 12.6% (95% CI: 12.3, 12.8) of the postpartum women used LARC methods. This is lower than the previous studies conducted in Kenya 38.3% [27], Rwanda 28.1% [28], and Ghana 28.6% [29]. Furthermore, it is lower than studies done in Ethiopia [6, 14, 30–32]. The possible justification might be due to the difference in the study setting. For example, the previous studies conducted in Kenya, Rwanda, Ghana, and Ethiopia were small-scale surveys compared with the DHS, which is a nationally representative survey that covered more women. It could also be due to socio-demographic differences, the proportion of postpartum women who had no formal education was more than 25% in the current study. However, almost all and more than 90% of postpartum women in the Kenya and Rwanda studies attended formal education, respectively. Education can increase women's awareness and level of understanding about the risk of being pregnant in the postpartum period [27, 32]. The other possible reason for the lower use of LARC methods in the current study could be the difference in place of delivery. In this study, 75.5% of postpartum women were delivered at health institutions, but almost all (98.4%) and all (100%) of postpartum women in Ghana and Rwanda were delivered at health institutions, respectively [28, 29]. Postpartum women who are delivered at health institutions are more likely to receive counseling about postpartum LARC methods [29, 33].

The current study revealed that the odds of using postpartum LARC was higher among postpartum women with secondary and higher education compared with those without formal education. This finding is in line with studies done in Ethiopia [34], Kenya [35], Uganda [36]. This might be due to educated women have easily access to information regarding the side effect and benefits of postpartum LARC methods [32]. They have a good understanding of the misconceptions and myths that often serve as a deterrent to the use of postpartum LARC [4]. Postpartum women who had media exposure to family planning messages were positively associated with LARC use. This result is in line with studies conducted in Uganda [37]. The possible explanation could be that postpartum women with media exposure might have a better understanding of LARC methods, which can lead to a positive change in their attitude toward LARC [38]. The study indicates that media exposure will reduce the barriers to access and use of maternal health care services, including FP.

**Table 3. Multi-level mixed-effect logistic regression analysis of factors associated with postpartum long acting reversible use (sub Saharan Africa, 2023).**

| Variables | Null model | Model I | Model II | Model III |
|---|---|---|---|---|
| Age of the respondents | | | | |
| 15–24 | | 1 | | 1 |
| 25–34 | | 1.27 (1.17, 1.40) | | 1.15 (0.98, 1.26) |
| 35–49 | | 0.91 (0.82, 1.12) | | 0.93 (0.82, 1.05) |
| Education of the mother | | | | |
| No formal education | | 1 | | 1 |
| Primary education | | 1.36 (1.25, 1.48) | | 1.51 (1.41, 1.53)[*] |
| Secondary education and above | | 1.21 (1.11, 1.31) | | 1.62 (1.32, 1.71)[*] |
| Wealth index | | | | |
| Poor | | 1 | | 1 |
| Middle | | 1.12 (1.03, 1.21) | | 1.21 (0.96, 1.53) |
| Rich | | 1.06 (0.96, 1.14) | | 1.25 (0.97, 1.38) |
| Media exposure | | | | |
| No | | 1 | | 1 |
| Yes | | 1.28 (1.19, 1.37) | | 1.73 (1.51, 1.85)[*] |
| Place of delivery | | | | |
| Home | | 1 | | 1 |
| Health institution | | 1.51 (1.38, 1.63) | | 1.54 (1.43, 1.67)[*] |
| Number of ANC visits | | | | |
| None | | 1 | | 1 |
| 1–3 | | 2.67 (2.35, 2.91) | | 2.62 (2.31, 2.83)[*] |
| ≥4 | | 3.34 (2.98, 3.563) | | 3.22 (2.93, 3.57)[*] |
| Received PNC | | | | |
| No | | 1 | | 1 |
| Yes | | 1.26 (1.13, 137) | | 1.34 (1.13, 1.58)[*] |
| Number of living children | | | | |
| None | | 1 | | 1 |
| 1–4 | | 1,55 (1.43, 1.69) | | 1.23 (0.89, 1.52) |
| ≥5 | | 1.58 (1.38, 1.75) | | 1.27(0.98, 1.59) |
| Wanted last-child | | | | |
| Wanted then | | 1 | | 1 |
| Wanted later | | 1.16 (1.05, 1.43) | | 1.12 (0.89, 1.28) |
| Wanted no more | | 1.23 (1.11, 1.48) | | 1.19 (0.95, 1.37) |
| Discuss with health professionals with the last 12 months | | | | |
| No | | 1 | | 1 |
| Yes | | 1.23 (1.17,1.29) | | 1.25 (0.98, 1.39) |
| Income level | | | | |
| Low income | | | 1 | 1 |
| Lower middle income | | | 2.01 (1.62, 2.33) | 2.41 (2.11, 2.88)[*] |
| Upper middle income | | | 1.51 (1.34, 1.87) | 1.83 (1.56, 2.24)[*] |
| Community media exposure | | | | |
| Low | | | 1 | 1 |
| High | | | 1.22 (1.12, 1.35) | 1.15 (0.97, 1.33) |
| Community level poverty | | | | |
| Low | | | 1.17 (0.95, 1.28) | 1.23 (0.96, 1.31) |
| High | | | 1 | 1 |
| Community level education | | | | |

*(Continued)*

**Table 3.** (Continued)

| Variables | Null model | Model I | Model II | Model III |
|---|---|---|---|---|
| Low | | | 1 | 1 |
| High | | | 1.21 (1.11, 1.36) | 1.11 (0.93, 1.25) |
| Random effect result | | | | |
| ICC | 41.6 | 39.6 | 37.8 | 34.5 |
| Variance | 16.6 | 15.5 | 14.1 | 13.2 |
| PCV (%) | Ref | 6.6 | 15.1 | 20.5 |
| Model fitness | | | | |
| LL | -29560.5 | -17689.2 | -29470.6 | -17675.2 |
| Deviance | 59121 | 35378.4 | 58941.2 | 35350.4 |

[*]P-value < 0.05, ICC; Intra class corrolation cofficent, PCV; proportional change in variance, LL; log-likelihood.

In this study, number of ANC visits was significantly associated with postpartum LARC use. The finding is consistent with studies conducted in Ethiopia [32], Mexico [39], and Nigeria [40]. Mothers who visit health institutions for ANC follow-up receive counseling about LARC that enhances their awareness, understanding, and appreciation of the benefits of postpartum LARC use and further promotes their practice [32, 40]. Moreover, mothers who had ANC follow-up during pregnancy were more likely to have an institutional delivery, which gives them the opportunity to receive postpartum LARC counseling. Thus, strengthening the existing effort to improve ANC follow-up and institutional delivery would help promote the utilization of immediate postpartum LARC [40].

Likewise, the odds of postpartum LARC use were higher among participants who had delivered at the health institutions as compared with mothers who had delivered at home. The finding is congruent with previous evidence from the Ethiopia [32, 41]. Women who had delivered at health institutions might possibly have a better opportunity to receive counseling related to postpartum LARC use immediately after delivery. That will further have a positive impact on the use of postpartum LARC use [33, 42]. Furthermore, women who had delivered at home were usually less educated and had less access to health messages, which might affect healthy practices, including modern contraceptive use [43]. Postpartum women who received PNC services were more likely to use LARC compared with their counterparts. The possible reason might be those postpartum women who received PNC services received contraceptive information and counseling before discharge, which can increase the subsequent uptake of LARC methods [44].

LARC use was more likely among postpartum women living in lower- and upper-middle-income countries than among those living in low-income countries. The reason might be that postpartum women from lower- and upper-middle-income countries are better able to deal with the cost barrier associated with access to FP use as compared to those from lower-income countries since they can be able to overcome both the direct and indirect costs associated with FP uptake [45]. Another possible reason could be that as income increases, exposure to different types of health information (media exposure) and the financial accessibility of services will improve [18, 45].

The main strength of the current study was that it used nationally representative survey data with large sample size. We employed multilevel analysis to accommodate the hierarchical nature of the data. However, it is impossible to infer cause-effect relationships between the dependent variable and the identified independent variables due to the cross-sectional nature of the study. Moreover, due to the use of secondary data, essential factors like fear of side

effects, attitude toward LARC methods, husband perspective on LARC methods, and sociocultural factors were not available in the DHS data set. Therefore, it was not possible to incorporate these variables in the analysis.

## Conclusion

Nearly one in 10 postpartum women used long-acting reversible contraceptives. This implied that most SSA postpartum women were exposed to unintended pregnancy and unsafe abortion. This raises the risk of maternal and child morbidity and mortality. Women's education, media exposure, place of delivery, number of ANC visits, received PNC, and income level were statistically significant factors for LARC use. Hence, we suggest that the concerned bodies should promote family planning messages in mass media (radio, newspapers, and television) and provide the well-documented benefits of postpartum LARC use. Promote the integration of postpartum LARC methods into maternal health care services. It is likely that such integration would help increase the use of LARC. This is because ANC visits, institutional delivery, and postnatal care services remain important windows of opportunity to provide access to LARC and offer women LARC methods. Give better attention to postpartum women living in low-income countries and uneducated women.

## Supporting information

**S1 Data.**
(CSV)

## Acknowledgments

Our gratitude is extended to the DHS programs for allowing us to use all of the relevant DHS data in this study.

## Author Contributions

**Conceptualization:** Desale Bihonegn Asmamaw.

**Data curation:** Tsegaw Amare.

**Formal analysis:** Desale Bihonegn Asmamaw, Tadele Biresaw Belachew,
 Atitegeb Abera Kidie.

**Funding acquisition:** Desale Bihonegn Asmamaw.

**Methodology:** Desale Bihonegn Asmamaw, Abel Endawkie.

**Software:** Desale Bihonegn Asmamaw, Sisay Maru Wubante.

**Supervision:** Desale Bihonegn Asmamaw.

**Validation:** Tadesse Tarik Tamir.

**Visualization:** Elsa Awoke Fentie.

**Writing – original draft:** Wubshet Debebe Negash.

**Writing – review & editing:** Samrawit Mihret Fetene, Banchlay Addis,
 Alebachew Ferede Zegeye, Elsa Awoke Fentie.

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
