## [Decision Letter · Decision Letter 0]

21 Jul 2023

PONE-D-23-11696Postpartum long-acting reversible contraceptives use in Sub-Saharan Africa. Evidence from recent Demographic and Health Surveys DataPLOS ONE

Dear Dr. Asmamaw,

Thank you for submitting your manuscript to PLOS ONE. After careful consideration, we feel that it has merit but does not fully meet PLOS ONE’s publication criteria as it currently stands. Therefore, we invite you to submit a revised version of the manuscript that addresses the points raised during the review process.

Please submit your revised manuscript by Sep 04 2023 11:59PM. If you will need more time than this to complete your revisions, please reply to this message or contact the journal office at plosone@plos.org. Please include the following items when submitting your revised manuscript:A rebuttal letter that responds to each point raised by the academic editor and reviewer(s). You should upload this letter as a separate file labeled 'Response to Reviewers'.A marked-up copy of your manuscript that highlights changes made to the original version. You should upload this as a separate file labeled 'Revised Manuscript with Track Changes'.An unmarked version of your revised paper without tracked changes. You should upload this as a separate file labeled 'Manuscript'.If applicable, we recommend that you deposit your laboratory protocols in protocols.io to enhance the reproducibility of your results. Protocols.io assigns your protocol its own identifier (DOI) so that it can be cited independently in the future. For instructions see: https://journals.plos.org/plosone/s/submission-guidelines#loc-laboratory-protocols. Additionally, PLOS ONE offers an option for publishing peer-reviewed Lab Protocol articles, which describe protocols hosted on protocols.io. Read more information on sharing protocols at https://plos.org/protocols?utm_medium=editorial-email&utm_source=authorletters&utm_campaign=protocols.

We look forward to receiving your revised manuscript.

Kind regards,

Sidrah Nausheen, FCPS

Academic Editor

PLOS ONE

Journal Requirements:

- https://doi.org/10.1186/s40834-022-00211-x

- https://doi.org/10.1371%2Fjournal.pone.0037905

- http://dx.doi.org/10.1186/s12889-023-15187-9

- https://doi.org/10.1186/s12889-020-09965-y

- https://doi.org/10.1186/s12905-022-01982-w

In your revision ensure you cite all your sources (including your own works), and quote or rephrase any duplicated text outside the methods section. Further consideration is dependent on these concerns being addressed.

3. Thank you for stating the following financial disclosure: "NA"

Reviewers' comments:

Reviewer's Responses to Questions

**Comments to the Author**

1. Is the manuscript technically sound, and do the data support the conclusions?

Reviewer #1: Yes

Reviewer #2: Yes

2. Has the statistical analysis been performed appropriately and rigorously? 

Reviewer #1: Yes

Reviewer #2: I Don't Know

3. Have the authors made all data underlying the findings in their manuscript fully available?

Reviewer #1: Yes

Reviewer #2: Yes

4. Is the manuscript presented in an intelligible fashion and written in standard English?

Reviewer #1: Yes

Reviewer #2: Yes

5. Review Comments to the Author

Reviewer #1: Comments to the Author:

This is an important study dealing with an absolutely relevant topic pertinent to reproductive health care.

Ethics Statement: We based this study on an analysis of existing survey data collected by the monitoring and evaluation. This is incomplete description to assess and use results of Demographic and Health Surveys (MEASURE DHS) project.

Abstract:

Methods:

Line 45: Secondary data analysis was performed using the recent Demographic and Health

Surveys (DHS).

Line 47: A multilevel mixed-effect logistic regression model was used to identify factors associated with long-acting reversible contraceptive use.

Introduction:

Line 92: Even though the postpartum period is the best opportunity for LARC insertion

Line 93- 95: Few studies have been conducted on LARC methods [4, 15, 16], however, majority of these focused on adolescent and reproductive-age women.

Methods:

Line 135: Of the community level factors, residence (rural, urban) were directly accessed from the EDHS data set. Full term required here for EDHS

Data Analysis

Specifications for models used are described clearly.

Results

Results are interpretated adequately.

Line 183: postpartum women also used LARC if they had institutional delivery.

Discussion

First paragraph should be reduced with minimum repetitions.

Conclusion

Appropriately summarizes the study findings.

Reviewer #2: This is a very well written article on the important topic of postpartum LARC use and factors that contribute to the use of LARC in Sub Saharan African countries.

The topic chosen is very important and addresses one of the essential strategies to meet the 2030 SGD of reducing MMR to below 70/100000.

The data has been taken from the DHS data base, thereby including a large population form different Sub Sahran countries with high MMR. Important factors like socioeconomic status, educational status, ANC visits PNC visits etc that can affect use of LARC have been touched upon.

The discussion is appropriately argued with respect to other studies.

Conclusion is aligned with the results.

6. PLOS authors have the option to publish the peer review history of their article (what does this mean?). If published, this will include your full peer review and any attached files.

Reviewer #1: No

Reviewer #2: **Yes: **Azra Amerjee

---

## [Author Response · Author response to Decision Letter 0]

30 Aug 2023

Dear reviewer, we would like to extend our deepest appreciation for devoting your time to reviewing our manuscript entitled " Postpartum long-acting reversible contraceptives use in Sub-Saharan Africa. Evidence from recent Demographic and Health Surveys Data". there has been a revision of this manuscript. The whole structure of the manuscript has been revised. We hope now the manuscript is clear and more acceptable than its previous version. We have tried to present the paper in the proper manner according to your comment on what to supposed to do so. For this, here we have given our responses to each of the concerns you raised. Again, we would like to remind our strongest gratitude for your effort in the improvement of this manuscript, and all the points were addressed in the point-by-point response.

Reviewer 1

Reviewer comments: Line 45: Secondary data analysis was performed using the recent Demographic and Health Survey (DHS)

Authors response: thank you for your observations and comments, this has been addressed, kindly see line 45.

Reviewer comments: Line 47: A multilevel mixed-effect logistic regression model was used to identify factors associated with long-acting reversible contraceptive use.

Authors response: Dear reviewer thank you for your comments, this has been addressed, kindly see line 46-47.

Reviewer comments: Line 92: Even though the postpartum period is the best opportunity for LARC insertion

Authors response: Dear reviewer, thank you for your comments, this has been addressed, kindly see line 92.

Reviewer comments: Line 93- 95: Few studies have been conducted on LARC methods [4, 15, 16], however, majority of these focused on adolescent and reproductive-age women.

Authors response: Thank you for your observations, this has been corrected accordingly, kindly see line 94-95.

Reviewer comments: Line 135: Of the community level factors, residence (rural, urban) were directly accessed from the EDHS data set. Full term required here for EDHS

Authors response: Thank you for your comments, this has been addressed, kindly see line 131-132.

Reviewer comments: Results are interpreted adequately.

Authors response: Thank you 

Reviewer comments: Line 183: postpartum women also used LARC if they had institutional delivery.

Authors response: Thank you for your suggestions, this has been addressed, kindly see line 185-186.

Reviewer comments: First paragraph should be reduced with minimum repetitions.

Authors response: thank you for your comments and suggestions, this has been corrected accordingly, kindly see line 215-217.

Reviewer 2

Reviewer comments: This is a very well written article on the important topic of postpartum LARC use and factors that contribute to the use of LARC in Sub Saharan African countries. The topic chosen is very important and addresses one of the essential strategies to meet the 2030 SGD of reducing MMR to below 70/100000. The data has been taken from the DHS data base, thereby including a large population form different Sub Sahran countries with high MMR. Important factors like socioeconomic status, educational status, ANC visits PNC visits etc that can affect use of LARC have been touched upon. The discussion is appropriately argued with respect to other studies.

Conclusion is aligned with the results.

Authors response: Thank you

---

## [Editor Report · Decision Letter 1]

1 Sep 2023

Postpartum long-acting reversible contraceptives use in Sub-Saharan Africa. Evidence from recent Demographic and Health Surveys Data

PONE-D-23-11696R1

Dear Desale Bihonegn Asmamaw,

We’re pleased to inform you that your manuscript has been judged scientifically suitable for publication and will be formally accepted for publication once it meets all outstanding technical requirements.

Kind regards,

Sidrah Nausheen, FCPS

Academic Editor

PLOS ONE
---

## [Editor Report · Acceptance letter]

26 Sep 2023

PONE-D-23-11696R1 

Postpartum long-acting reversible contraceptives use in sub-Saharan Africa. Evidence from recent Demographic and Health Surveys Data 

Dear Dr. Asmamaw:

I'm pleased to inform you that your manuscript has been deemed suitable for publication in PLOS ONE. Congratulations! Your manuscript is now with our production department. 

Kind regards, 

on behalf of

Dr. Sidrah Nausheen 

Academic Editor

PLOS ONE